# D²P-FED: Differentially Private Federated Learning with Efficient Communication

## Abstract

In this paper, we propose the **d**iscrete Gaussian based **d**ifferentially **p**rivate **fed**erated learning (D²P-FED), a unified scheme to achieve both differential privacy (DP) and communication efficiency in federated learning (FL). In particular, compared with the only prior work taking care of both aspects, D²P-FED provides stronger privacy guarantee, better composability and smaller communication cost. The key idea is to apply the discrete Gaussian noise to the private data transmission. We provide complete analysis of the privacy guarantee, communication cost and convergence rate of D²P-FED. We evaluated D²P-FED on INFIMNIST and CIFAR10. The results show that D²P-FED outperforms the-state-of-the-art by 4.7% to 13.0% in terms of model accuracy while saving one third of the communication cost. The results might be surprising at its first glance but is reasonable because the quantization level $k$ in D²P-FED is independent of $q$. As long as $q$ is large enough, the probability that the noise exceeds $q$ is small and thus has negligible impact on the model accuracy.

## 1 Introduction

Federated learning (FL) is a popular machine learning paradigm that allows a central server to train models over decentralized data sources. In federated learning, each client performs training locally on their data source and only updates the model change to the server, which then updates the global model based on the aggregated local updates. Since the data stays locally, FL can provide better privacy protection than traditional centralized learning. However, FL is facing two main challenges: (1) FL lacks a rigorous privacy guarantee (e.g., differential privacy (DP)) and indeed, it has been shown to be vulnerable to various inference attacks (Nasr et al., 2019; Pustozerova & Mayer; Xie et al., 2019); (2) FL incurs considerable communication costs. In many potential applications of FL such as mobile devices, these two challenges are present *simultaneously*.

However, privacy and communication-efficiency have mostly been studied independently in the past. As regards privacy, existing work has applied a gold-standard privacy notion – differential privacy (DP) – to FL, which ensures that the server could hardly determine the participation of each client by observing their updates (Geyer et al., 2017). To achieve DP, each client needs to inject noise to their local updates and as a side effect, the performance of the trained model would inevitably degrade. To improve model utility, secure multiparty computation (SMC) has been used in tandem with DP to reduce noise (Jayaraman et al., 2018; Truex et al., 2019). The key idea is to prevent the server from observing the individual updates, make only the aggregate accessible, and thus transform from local DP to central DP. However, SMC introduces extra communication overhead to each client. There has been extensive research on improving communication efficiency of FL while ignoring the privacy aspect (Tsitsiklis & Luo, 1987; Balcan et al., 2012; Zhang et al., 2013; Arjevani & Shamir, 2015; Chen et al., 2016). However, these communication reduction methods either have incompatible implementations with the existing DP mechanisms or would break the DP guarantees when combined with SMC.

The only existing work that tries to reconcile DP and communication efficiency in FL is cpSGD (Agarwal et al., 2018). The authors leveraged the Binomial mechanism, which adds Binomial noise into local updates to ensure differential privacy. The discrete nature of Binomial noise allows it to be transmitted efficiently. However, cpSGD faces several limitations when applied to real-world applications. Firstly, with Binomial noise, the output of a learning algorithm would have different supports on different input datasets; as a result, Binomial noise can only guarantee approx-

imate DP where the participation of the client can be completely exposed with nonzero probability. Also, there lacks a tight composition for DP with Binomial noise and the resulting privacy budget skyrockets in a multi-round FL protocol. Hence, the Binomial mechanism cannot produce a useful model with a reasonable privacy budget on complex tasks. Last but not least, the Binomial mechanism involves several mutually constrained hyper-parameters and the privacy formula is extremely complicated, which makes hyper-parameter tuning a difficult task.

In this paper, we propose the **d**iscrete Gaussian based **d**ifferential **p**rivate **fed**erated learning ($D^2$P-FED), an alternative technique to reduce communication costs while maintaining differential privacy in FL. Our key idea is to leverage the discrete Gaussian mechanism in FL, which adds discrete Gaussian noise into client updates. We show that the discrete Gaussian mechanism satisfies Rényi DP which provides better composability. We employ secure aggregation along with the discrete Gaussian mechanism to lower the noise and exhibit the privacy guarantee for this hybrid privacy protection approach. To save the communication cost, we integrate the stochastic quantization and random rotation into the protocol. We then cast FL as a general distributed mean estimation problem and provide the analysis of the utility for the overall protocol. Our theoretical analysis sheds light on the superiority of $D^2$P-FED to cpSGD. Our experiments show that $D^2$P-FED can lead to state-of-the-art performance in terms of managing the trade-off among privacy, utility, and communication.

## 2 RELATED WORK

It is well studied how to improve the communication cost in traditional distributed learning settings (Tsitsiklis & Luo (1987); Balcan et al. (2012); Zhang et al. (2013); Arjevani & Shamir (2015); Chen et al. (2016)). However, most of the approaches either require communication between the workers or are designed for specific learning tasks so they cannot be applied directly to general-purpose FL. The most relevant work is Suresh et al. (2017) which proposed to use stochastic quantization to save the communication cost and random rotation to lower mean squared error of the estimated mean. We follow their approach to improve the communication efficiency and model utility of $D^2$P-FED. Nevertheless, our work differs from theirs in that we also study how to ensure DP for rotated and quantized data transmission and prove a convergence result for the learning algorithm with both communication cost reduction and privacy protection steps in place.

On the other hand, differentially private FL is undergoing rapid development during the past few years (Geyer et al. (2017); McMahan et al. (2017); Jayaraman et al. (2018)). However, these methods mainly focus on improving utility under a small privacy budget and ignore the issue of communication cost. In particular, we adopt a similar hybrid approach to Truex et al. (2019), which combines SMC with DP for reducing the noise. SMC ensures that the centralized server can only see the aggregated update but not individual ones from clients and as a result, the noise added by each client can be reduced by a factor of the number of clients participating in one round. The difference of our work from theirs is that we inject discrete Gaussian noise to local updates instead of the continuous Gaussian noise. This allows us to use secure aggregation (Bonawitz et al., 2017) which is much cheaper than threshold homomorphic encryption used by Truex et al. (2019). We further study the interaction between discrete Gaussian noise and the secure aggregation as well as their effects on the learning convergence.

We identify cpSGD (Agarwal et al. (2018)) as the most comparable work to $D^2$P-FED. Just like $D^2$P-FED, cpSGD aims to improve both the communication cost and the utility under rigorous privacy guarantee. However, cpSGD suffers from three main defects discussed in Section 1. This paper proposes to use the discrete Gaussian mechanism to mitigate these issues in cpSGD.

## 3 BACKGROUND AND NOTATION

In this section, we provide an overview of FL and DP and establish the notation system. We use bold lower-case letters (*e.g.* **a**,**b**,**c**) to denote vectors, and bold upper-case letters (*e.g.* **A**, **B**, **C**) for matrices. We denote $1 \cdots n$ by $[n]$.

**FL Overview.** In a FL system, there are one server and $n$ clients $\mathcal{C}_i, i \in [n]$. The server holds a global model of dimension $d$. Each client holds (IID or non-IID) samples drawn from some unknown distribution $\mathcal{D}$. The goal is to learn the global model $\boldsymbol{w} \in \mathbb{R}^d$ that minimizes some loss function $\mathcal{L}(\boldsymbol{w}, \mathcal{D})$. To achieve this, the system runs a $T$-round FL protocol. The server initializes

the global model with $\boldsymbol{w}_0$. In round $t \in [T]$, the server randomly sub-samples $\gamma n$ clients from $[n]$ with sub-sampling rate $\gamma$ and broadcasts the global model $\boldsymbol{w}_{t-1}$ to the chosen clients. Each chosen client $\mathcal{C}_i$ then runs the local optimizers (*e.g.* SGD, Adam, and RMSprop), computes the difference between the locally optimized model $\boldsymbol{w}_t^{(i)}$ and the global model $\boldsymbol{w}_{t-1}$: $\boldsymbol{g}_t^{(i)} = \boldsymbol{w}_t^{(i)} - \boldsymbol{w}_{t-1}$, and uploads $\boldsymbol{g}_t^{(i)}$ to the server. The server takes the average of the differences and update the global model $\boldsymbol{w}_t = \boldsymbol{w}_{t-1} + \frac{1}{k} \sum \boldsymbol{g}_t^{(i)}$.

**Communication in FL.** The clients in FL are often edge devices, where the upload bandwidth is fairly limited; therefore, communication efficiency is of uttermost importance to FL. Let $\pi$ denote a communication protocol. We denote the per-round communication cost as $\mathcal{C}(\pi, \boldsymbol{g}^{[n]})$. To lower the communication cost, the difference vectors are typically compressed before sent to the server. The compression would degrade model performance and we measure the performance loss via the mean squared error. Specifically, letting $\bar{\boldsymbol{g}}$ denote the actual mean of difference vectors $\frac{1}{n} \sum_{i=1}^{n} \boldsymbol{g}^{(i)}$ and $\tilde{\boldsymbol{g}}$ denote the server's estimated mean of difference vectors using some protocol such as $\mathrm{D}^2\mathrm{P}$-FED, we could measure the performance loss by $\mathcal{E}(\pi, \boldsymbol{g}^{[n]}) = \mathbb{E}[\|\tilde{\boldsymbol{g}} - \bar{\boldsymbol{g}}\|^2]$, i.e., the mean squared error between the estimated and the actual mean. This mean squared error is directly related to the convergence rate of FL (Agarwal et al., 2018).

**Threat Model & Differential Privacy.** We assume that the server is honest-but-curious. Namely, the server will follow the protocol honestly under the law enforcement or reputation pressure, but is curious to learn the client-side data from the legitimate client-side messages. In the FL context, the server wants to get information about the client-side data by studying the local updates received without deviating from the protocol.

The above attack, widely known as the inference attack (Shokri et al., 2017; Yeom et al., 2018; Nasr et al., 2019), can be effectively mitigated using a canonical privacy notation namely differential privacy (DP). Intuitively, DP, in the context of ML, ensures that the trained model is nearly the same regardless of the participation of any arbitrary client.

**Definition 1** (($\epsilon, \delta$)-DP). *A randomized algorithm $f : \mathcal{D} \to \mathcal{R}$ is ($\epsilon, \delta$)-differentially private if for every pair of neighboring datasets $D$ and $D'$ that differs only by one datapoint, and every possible (measurable) output set $E$ the following inequality holds: $P[f(D) \subseteq E] \leq e^{\epsilon} P[f(D') \subseteq E] + \delta$.*

($\epsilon, \delta$)-DP has been used as a privacy notion in most of the existing works of privacy-preserving FL. However, in this paper, we consider a generalization of DP, Rényi differential privacy (RDP), which is strictly stronger than ($\epsilon, \delta$)-DP for $\delta > 0$ and allows tighter analysis for compositing multiple mechanisms. This second point is particularly appealing, as FL mostly comprises multiple rounds yet the existing works suffer from skyrocketing privacy budgets for multi-round learning.

**Definition 2** (($\alpha, \epsilon$)-RDP). *For two probability distributions $P$ and $Q$ with the same support, the Rényi divergence of order $\alpha > 1$ is defined by $D_\alpha(P\|Q) \triangleq \frac{1}{\alpha-1} \log \mathbb{E}_{x \sim Q}(\frac{P(x)}{Q(x)})^\alpha$. A randomized mechanism $f : \mathcal{D} \to \mathcal{R}$ is ($\alpha, \epsilon$)-RDP, if for any neighboring datasets $D, D' \in \mathcal{D}$ it holds that $D_\alpha(f(D)\|f(D')) \leq \epsilon$.*

The intuition behind RDP is the same as other variants of differential privacy: "Similar inputs should yield similar output distributions," and the similarity is measured by the Rényi divergence under RDP. RDP can also be converted to ($\epsilon, \delta$)-DP using the following transformation.

**Lemma 1** (RDP-DP conversion (Mironov (2017))). *If $\mathcal{M}$ obeys ($\alpha, \epsilon$)-RDP, then $\mathcal{M}$ obeys ($\epsilon + \log(1/\delta)/(\alpha - 1), \delta$)-DP for all $0 < \delta < 1$.*

RDP enjoys an operationally convenient and quantitatively accurate way of tracking cumulative privacy loss when compositing multiple mechanisms (Lemma 2) or being combined with subsampling (Wang et al., 2018). As a result, RDP is particularly suitable for the context of ML.

**Lemma 2** (Adaptive composition of RDP (Mironov (2017))). *If (randomized) mechanism $\mathcal{M}_1$ obeys ($\alpha, \epsilon_1$)-RDP, and $\mathcal{M}_2$ obeys ($\alpha, \epsilon_2$)-RDP, then their composition obeys ($\alpha, \epsilon_1 + \epsilon_2$)-RDP.*

## 4 DISCRETE GAUSSIAN MECHANISM

In this section, we present the discrete Gaussian mechanism and establish its privacy guarantee. We first introduce discrete Gaussian distribution.

**Definition 3** (Discrete Gaussian Distribution). *Discrete Gaussian is a probability distribution on a discrete additive subgroup $\mathbb{L}$ (for instance, a multiple of $\mathbb{Z}$) parameterized by $\sigma$. For a discrete Gaussian distribution $N_{\mathbb{L}}(\sigma)$ and $x \in \mathbb{L}$, the probability mass on $x$ is proportional to $e^{-x^2/(2\sigma^2)}$.*

Discrete Gaussian mechanism works by adding noise drawn from discrete Gaussian distribution. Canonne et al. (2020) proved concentrated DP for the discrete Gaussian mechanism. However, there lacks tight privacy amplification and composition theorem for concentrated DP. To address, we turn to RDP and provide the first RDP analysis for the discrete Gaussian mechanism. The proof is delayed to Appendix A due to space limitation.

**Theorem 1** (RDP for discrete Gaussian mechanism). *If $f$ has sensitivity $1$ and $range(f) \subseteq \mathbb{L}$, then the discrete Gaussian mechanism: $f(\cdot) + N_{\mathbb{L}}(\sigma)$ satisfies $(\alpha, \alpha/(2\sigma^2))$-RDP.*

Under RDP, discrete Gaussian exhibits tight privacy amplification bound under sub-sampling (Wang et al., 2018). This suits FL well since a subset of clients is sub-sampled to upload the updates in each round.

**Corollary 1** (Privacy amplification for discrete Gaussian mechanism (Wang et al., 2018)). *If a discrete Gaussian mechanism is $(\alpha, \frac{\alpha}{2\sigma^2})$-RDP, then augmented with subsampling (without replacement), the privacy guarantee is amplified to (1) $(\alpha, \mathcal{O}(\frac{\alpha\gamma^2}{\sigma^2}))$ in the high privacy regime; or (2) $(\alpha, \mathcal{O}(\alpha\gamma^2 e^{\frac{1}{\sigma^2}}))$ in the low privacy regime.*

Besides, RDP enables discrete Gaussian mechanism to be composed tightly with analytical moments accountant (Wang et al., 2018), which saves a huge amount of privacy budget in a multi-round FL. Analytical moments accountant is a data structure that tracks the cumulant generating function of the composed mechanisms symbolically. Since it has no closed-form solution, we instead introduce the canonical composition of RDP (Mironov, 2017) below for the ease of discussion in Section 6.3.

**Corollary 2** (Composition for discrete Gaussian mechanism (Wang et al., 2018)). *If a discrete Gaussian mechanism is $(\alpha, \frac{\alpha}{2\sigma^2})$-RDP, then the sequential composition of $T$ such mechanisms yield $(\alpha, \frac{T\alpha}{2\sigma^2})$-RDP guarantee. If we convert all the RDP guarantees back to $(\epsilon, \delta)$-DP, the growth of $\epsilon$ under the same $\delta$ is asymptotically $\mathcal{O}(\sqrt{T})$.*

Note that both privacy amplification and composition are given in an asymptotic form for the clarity of presentation. For tight bound, we refer the readers to Theorem 27 and Section 3.3 in Wang et al. (2018).

## 5 D²P-FED: ALGORITHM AND PRIVACY ANALYSIS

In this section, we formally present D²P-FED and provide rigorous privacy analysis.

### 5.1 ALGORITHM

Algorithm 1 provides the pseudocode for D²P-FED. It follows the general FL pipeline which iteratively performs the following steps: (1) the server broadcasts the global model to a subset of clients; (2) the selected clients train the global model on their local data and upload the resulting model difference; and (3) the server aggregates the model differences uploaded by the clients and updates the global model. Grounded on the general FL pipeline, D²P-FED introduces some additional steps at the client side as follows to improve communication efficiency and privacy.

**Stochastic Quantization & Random Rotation (line 11-14).** To lower the communication cost, the clients stochastically quantize the values in the update vectors to some discrete domain. Compared with real number encoding which costs 32 or 64 bits per dimension, the quantized value only requires $\log_2(k)$ bits per dimension where $k$ is the level of quantization (see line 12-14 in Algorithm 1 and McMahan et al. (2016) for detailed explanation.). On the other hand, quantization lowers the fidelity of the update vector and thus leads to some error in estimating the mean of gradients. To lower the estimation error, the clients apply randomized rotation to the updates before quantization as proposed by McMahan et al. (2016). The details are discussed in Section 6.1.

**Discrete Gaussian Mechanism (line 15).** We apply the discrete Gaussian mechanism to ensure DP. To determine the noise magnitude to be added, we need to bound the $\ell_2$-sensitivity (defined in Section 5.2) of the gradient aggregate. Without quantization and random rotation, one could clip the individual gradient update and consequently the $\ell_2$-sensitivity is just the clipping threshold. However, the inclusion of compression steps makes the analysis of $\ell_2$-sensitivity more sophisticated. We provide the analysis of sensitivity and RDP guarantees for the entire algorithm in Section 5.2. Each client samples the noise from the discrete Gaussian distribution. In contrast to prior work where each client adds independent noise, we require the clients to share the same random seed, generate the same noise and add an average share of the noise in each round. This is because the sum of multiple independent discrete Gaussians is no longer a discrete Gaussian. Note that the same random seed is also required for communication reduction in secure aggregation so we can conveniently reuse it here without introducing more further overhead (see Bonawitz et al. (2017) for details). The noise magnitude is set to ensure that the aggregate noise from all clients provides the global DP.

---

**Algorithm 1:** D²P-FED Protocol.

---

**Input:** Support Lattice: $\mathbb{L} = \frac{2\boldsymbol{g}^{\max}}{k-1} \cdot \mathbb{Z}$, Noise Scale: $\sigma$, Rotation Matrix: $\boldsymbol{R}$, Random Seed: s
Quantization Level: $k$, $\phi_q(x) = \frac{2\boldsymbol{g}^{\max}}{k-1}((\frac{k-1}{2\boldsymbol{g}^{\max}}x + \frac{q-1}{2}) \mod q - \frac{q-1}{2})$, $q$ is odd.

1   **for** $t \leftarrow [T]$ **do**
2    **Server:**
3     Sub-sample a subset of clients $S \subset [n], |S| = \gamma n$ and broadcast $\boldsymbol{w}_{t-1}$ and $\boldsymbol{g}^{\max}$ to $S$
4    **Client:**
5     **foreach** *client* $i \in S$ **do** Send $\boldsymbol{u}_{ij}$ to $j$, $\boldsymbol{u}_{ij} \sim Unif(\mathbb{L}_q^d), \mathbb{L}_q := \{x \in \mathbb{L}, |x| \le \frac{q-1}{2}\}$
6    **Client:**
7     **foreach** *client* $i \in S$ **do**
8      Train the model $\boldsymbol{w}_t^{(i)}$ with $\boldsymbol{w}_t$ as initialization
9      $\boldsymbol{g}_t^{(i)} = \mathbf{w}_t^{(i)} - \mathbf{w}_{t-1}$        /* **Compute the difference** */
10      $\boldsymbol{g}_{t,clipped}^{(i)} = \boldsymbol{g}_t^{(i)} / \max(1, \frac{\|\boldsymbol{g}_t^{(i)}\|_2}{D})$      /* **Clip the difference** */
11      $\boldsymbol{g}_{t,rotated}^{(i)} = \boldsymbol{R} \times \boldsymbol{g}_{t,clipped}^{(i)}$      /* **Random Rotation** */
12      let $\boldsymbol{b}[r] := -\boldsymbol{g}^{\max} + \frac{2r\boldsymbol{g}^{\max}}{k-1}$ for every $r \in [0, k)$    /* **Quantize** */
13      **for** $j \in d, \boldsymbol{b}[r] \le \tilde{\boldsymbol{g}}_{t,quantized}^{(i)}[j] \le \boldsymbol{b}[r+1]$ **do**
14       $\tilde{\boldsymbol{g}}_{t,quantized}^{(i)}[j] = \begin{cases} \boldsymbol{b}[r+1] & \text{w.p. } \dfrac{\boldsymbol{g}_{t,rotated}^{(i)}[j] - \boldsymbol{b}[r]}{\boldsymbol{b}[r+1] - \boldsymbol{b}[r]} \\ \boldsymbol{b}[r] & \text{o.w.} \end{cases}$
15      $\tilde{\boldsymbol{g}}_{t,dp}^{(i)} = \tilde{\boldsymbol{g}}_{t,quantized}^{(i)} + \frac{\boldsymbol{\nu}_i}{\gamma n}, \boldsymbol{\nu}_i \overset{s}{\sim} N_{\mathbb{L}}^d(\sigma)$    /* **Discrete Gaussian** */
16      $\tilde{\boldsymbol{g}}_t^{(i)} = \phi_q(\phi_q(\tilde{\boldsymbol{g}}_{t,dp}^{(i)}) + \sum_{j \ne i, j \in S} \boldsymbol{u}_{ij} - \sum_{j \ne i, j \in S} \boldsymbol{u}_{ji})$    /* **Mask** */
17      Send $\tilde{\boldsymbol{g}}_t^{(i)}$ to the server
18    **Server:**
19     $\tilde{\boldsymbol{g}}_t = \frac{1}{\gamma n} \sum_{i \in S} \tilde{\boldsymbol{g}}_t^{(i)}$         /* **Aggregate** */
20     $\boldsymbol{w}_t = \boldsymbol{w}_{t-1} + \tilde{\boldsymbol{g}}_t$

---

**Secure aggregation. (line 5,16,19)** To reduce the noise magnitude for ensuring DP, we hide clients' individual updates from the central server and only allow it to see the aggregated updates via the technique of secure aggregation. If the individual updates were available to the central server, they should also be protected with the same privacy guarantee as the averaged update; in this case, the required noise scales up with $\mathcal{O}(\gamma n)$. On the other hand, if the central server can only access the aggregated updates, then the required noise is $\mathcal{O}(1)$. Hence, secure aggregation of local updates can lead to significant noise reduction. However, there exists a challenge for integrating secure aggregation with discrete Gaussian mechanism: discrete Gaussian variables have infinite support, thereby incompatible with secure aggregation which operates on finite field. In Section 5.2, we propose to address this challenge by mapping the noised vector to a cyclic additive group and then applying secure aggregation and show that the RDP guarantees are preserved under the mapping.

## 5.2 PRIVACY ANALYSIS

Before applying discrete Gaussian mechanism to D²P-FED, we need to figure out how to calibrate the added noise. In differential privacy, the calibration is guided by *sensitivity* of the function as defined below.

**Definition 4** ($\ell_2$-sensitivity). *Given a function $f : \mathcal{D} \to \mathbb{R}$ and two neighboring datasets $D$ and $D'$, the $\ell_2$-sensitivity of $f$: $\Delta_f$ is defined as $\Delta_f = \max_{D,D'} \|f(D) - f(D')\|_2$.*

In DP for deep learning, the traditional way to bound $\ell_2$-sensitivity is to clip the update vector. However, quantization will further influence the sensitivity after clipping. We provide the sensitivity after quantization as below. The proof is delayed to Appendix B due to space limitation.

**Theorem 2.** *If we clip the $l_2$ norm of $\boldsymbol{g}$ to $D$, and quantize it to $k = \sqrt{d} + 1$ levels, then the $\ell_2$ sensitivity of the difference is $4D$.*

Given Theorem 1 and Theorem 2, we provide the RDP bound for D²P-FED.

**Corollary 3** (RDP for D²P-FED). *Given the clipping bound $D$, the noise scale $\sigma$, D²P-FED follows $(\alpha, \frac{8\alpha D^2}{\sigma^2})$-RDP.*

**Remark 1: Comparison with cpSGD**    It seems unclear how to interpret the above bound when compared with Theorem 1 in cpSGD (Agarwal et al., 2018). Indeed, the claim that D²P-FED has a better privacy guarantee than cpSGD can be mainly justified by the following three aspects: (1) D²P-FED follows RDP which is a strictly stronger privacy notion than cpSGD which is intrisically limited to $(\epsilon, \delta)$-DP; (2) D²P-FED enjoys a tighter composition compared to cpSGD. This is of critical significance in a FL protocol with potentially thousands of rounds; (3) Our experimental results in Figure 1a also empirically show that D²P-FED enjoys a tighter composition than cpSGD. The total privacy budget for D²P-FED grows much more slowly than cpSGD as training proceeds.

**Remark 2: Privacy Effect of Secure Aggregation.**    Corollary 3 is built on the assumption that the centralized server only has access to the summed updates but not the individual ones. If the centralized server has access to individual updates, the noise has to scale up $\gamma n$ times to maintain the same privacy guarantee which severely hinders the model accuracy. To consolidate the assumption, we leverage a cryptographic technique, secure aggregation (Bonawitz et al., 2017) which guarantees that the centralized server can only see the aggregated result. The basic intuition is to mask the inputs with random values canceling out in pairs. However, since discrete Gaussian has infinite support, we cannot directly apply random masks to it. To reconcile secure aggregation with discrete Gaussian, we propose to project the involved values into a quotient group after shifting and then apply the random masks as shown in line 16 in Algorithm 1. According to the post-processing theorem of RDP (Mironov (2017)), the result still follows rigorous Rényi differential privacy as proved in Appendix C. Note that we consider a simplified version of the full secure aggregation protocol (Bonawitz et al. (2017)) in Algorithm 1 and omit many interesting details such as the generation of the random masks and how to deal with dropout. We deem this to be enough to clarify the idea behind the reconciliation. The complete version of secure aggregation can be reconciled using exactly the same trick.

**Theorem 3** (Informal). *Distributed discrete Gaussian mechanism with secure aggregation obeys the same RDP guarantee as vanilla global discrete Gaussian mechanism with the same parameters.*

## 6 COMMUNICATION PROTOCOL & UTILITY ANALYSIS

In this section, we present our communication protocol in detail and discuss the communication cost and the estimation error of D²P-FED with direct comparison to cpSGD. The drastic improvement of D²P-FED mainly comes from the tight composition of discrete Gaussian mechanism compared to binomial mechanism in cpSGD.

### 6.1 COMMUNICATION PROTOCOL

As the first step, we leverage stochastic $k$-level quantization proposed by McMahan et al. (2016) to lower the communication cost as described in line 12-14 in Algorithm 1. If we denote vanilla

stochastic k-level quantization with $\pi_k$, then we successfully reduce the per-round communication cost $\mathcal{C}(\pi_k, \boldsymbol{g}^{[n]}) = n \cdot (d\lceil \log_2 k \rceil + \tilde{\mathcal{O}}(1))$.

However, stochastic quantization sacrifices some accuracy for communication efficiency. Concretely, $\mathcal{E}(\pi_k, \boldsymbol{g}^{[n]}) = \mathcal{O}(\frac{d}{n} \cdot \frac{1}{n} \sum_{i=1}^{n} \|\boldsymbol{g}^{(i)}\|_2^2)$. Since the dimension of parameters $d$ is tens of thousands to hundreds of thousands in federated learning, the estimation error of the mean is too large. Thus to reduce the estimation error, we randomly rotate the difference vector (McMahan et al., 2016) as the second step. The key intuition is that the MSE of stochastic uniform quantization is $\mathcal{O}(\frac{d}{n}(\boldsymbol{g}^{\max})^2)$. With random rotation, we can limit $\boldsymbol{g}^{\max}$ to $\sqrt{\frac{\log d}{d}}$ w.h.p. so the MSE will be improved to $\mathcal{O}(\frac{\log d}{n})$. Agarwal et al. (2018) also leverages random rotation to reduce MSE. However, in their setting, random rotation intrinsically harms their privacy guarantee because the $\ell_\infty$-sensitivity might increase with rotation. A natural advantage of discrete Gaussian is that the privacy guarantee only depends on $\ell_2$-sensitivity which is an invariant under rotation. Thus, random rotation does not harm our privacy guarantee at all. We omit the details here and refer the interested readers to McMahan et al. (2016). We denote the protocol using $k$-level quantization and random rotation with $\pi_k^{(rot)}$. We know that $\mathcal{C}(\pi_k^{(rot)}, \boldsymbol{g}^{[n]})$ remains the same while the MSE error is reduced to $\mathcal{E}(\pi_k^{(rot)}, \boldsymbol{g}^{[n]}) = \mathcal{O}(\frac{\log d}{n} \cdot \frac{1}{n} \sum_{i=1}^{n} \|\boldsymbol{g}^{(i)}\|_2^2)$.

## 6.2 CONVERGENCE RATE OF D²P-FED

In this section, we relate the convergence rate with mean squared error using Corollary 4 and analyze the mean squared error of mean estimation in D²P-FED. Note that we assume each client executes one iteration in each round so $\boldsymbol{g}$ equals to the gradient or belongs to the sub-gradients.

**Corollary 4** (Ghadimi & Lan (2013)). *Let $F(\boldsymbol{w}) = \mathcal{L}(\boldsymbol{w}, \mathcal{D})$ for some given distribution $\mathcal{D}$. Let $F(\boldsymbol{w})$ be $L$-smooth and $\forall \boldsymbol{w}, \|\nabla F(\boldsymbol{w})\| \leq \rho$. Let $\boldsymbol{w}^0$ satisfy $F(\boldsymbol{w}^0) - F(\boldsymbol{w}^*) \leq \rho_F$. Then after $T$ rounds*

$$\mathbb{E}_{t \sim (Unif(T))}[\|\nabla F(\boldsymbol{w}_t)\|_2^2] \leq \frac{2\rho_F L}{T} + \frac{2\sqrt{2}\lambda\sqrt{L\rho_F}}{\sqrt{T}} + \rho B$$

*, where $\lambda^2 = \max_{1 \leq t \leq T} 2\mathbb{E}[\|\boldsymbol{g}(\boldsymbol{w}_t) - \nabla F(\boldsymbol{w}_t)\|_2^2] + 2\max_{1 \leq t \leq T} \mathbb{E}_q[\|\boldsymbol{g}(\boldsymbol{w}_t) - \tilde{\boldsymbol{g}}(\boldsymbol{w}_t)\|_2^2]$, and $B = \max_{1 \leq t \leq T} \|\boldsymbol{g}(\boldsymbol{w}_t) - \tilde{\boldsymbol{g}}(\boldsymbol{w}_t)\|$.*

As corollary 4 indicates, for a given gradient bound, the convergence rate approximately grows with the reciprocal of MSE. Thus, we analyze D²P-FED's MSE and obtain the following theorem. The proof is delayed to Appendix D due to space limitation.

**Theorem 4.** *If we choose $\sigma \geq 1/\sqrt{2\pi}$, the mean squared error is*

$$\mathcal{E}(\pi_{k,q,N_{\mathbb{L}}(\sigma^2)}^{(rot)}, \boldsymbol{g}^{[n]}) \leq (1 - \frac{1}{1 + 3e^{-2\pi^2\sigma^2}}(1 - \Phi(nq))) \cdot \frac{4d(\boldsymbol{g}^{\max})^2}{n(k-1)^2}(\frac{1}{4} + \frac{\sigma^2}{\gamma^2 n^2}) + (1 - \Phi(n(q-k-1))) \cdot q^2$$

*where $\Phi$ is the cumulative distribution function (CDF) of the standard normal distribution.*

**Remark 1: Choice of $\boldsymbol{g}^{\max}$.** As indicated by Theorem 4, the dominant term in MSE is proportional to the square of $\boldsymbol{g}^{\max}$. A natural choice of $\boldsymbol{g}^{\max}$ is the clipping bound $D$. If we want to match up with the MSE guarantee in cpSGD: $\mathcal{O}(\frac{\sigma^2 \log(d)}{n(k-1)^2})$, we need to inherit their choice of $\boldsymbol{g}^{\max} = \mathcal{O}(D\sqrt{\frac{\log(d)}{d}})$, and this can be achieved by clipping $l_\infty$ norm of the gradient after random rotation. For instance, according to Lemma 8 in Agarwal et al. (2018), we can choose $\boldsymbol{g}^{\max} = \frac{2\sqrt{\log(\frac{2nd}{\delta})}D}{\sqrt{d}}$. In that case, the possibility that the maximum of $\boldsymbol{g}$ exceeds $\boldsymbol{g}^{\max}$ is at most $\delta$. It follows that the possibility that the $\ell_\infty$-clipping really changes the update is bounded by $\delta$. Hence, the RHS of the MSE bound in Theorem 4 evolves to $(1 - \frac{1}{1 + 3e^{-2\pi^2\sigma^2}}(1 - \Phi(nq)) - \delta) \cdot \frac{4d(\boldsymbol{g}^{\max})^2}{n(k-1)^2}(\frac{1}{4} + \frac{\sigma^2}{\gamma^2 n^2}) + (1 - \Phi(n(q-k-1)) + \delta) \cdot q^2$, which is on the same order with the original bound given $\delta$ is small.

**Remark 2: Comparison with cpSGD.** As the MSE bound is on the same order with cpSGD (even the constants are close!), a natural question is "what is the advantage of D²P-FED over cpSGD in

terms of MSE?" Indeed, the advantage stems from a smaller standard deviation of the noise. Given a fixed privacy budget, due to the tighter composition of sub-sampled RDP (Wang et al., 2018), each round of D²P-FED can have more privacy budget and thus smaller noise scale. Plugging a smaller $\sigma$ into Theorem 4 will give a better MSE and thus a better convergence rate as cpSGD follows a similar convergence rate bound. Moreover, according to Figure 1 in Agarwal et al. (2018), even with the same noise scale, Gaussian noise provides stronger privacy guarantee than Binomial noise. As discrete Gaussian noise follows the same RDP bound as Gaussian noise, we believe discrete Gaussian can map the same-scale noise to lower privacy cost.

### 6.3 COMMUNICATION COST OF D²P-FED

First, we provide the trivial per-round communication cost which is exactly the same as cpSGD.

**Theorem 5.** *The per-round communication cost of* D²P-FED *is*

$$\mathcal{C}(\pi_{k,q,N_{\mathbb{L}}(\sigma^2)}^{(rot)}, \boldsymbol{g}^{[n]}) = n \cdot (d \log(nq + 1) + \tilde{\mathcal{O}}(1))$$

.

Now let's compare the number of rounds in cpSGD and D²P-FED qualitatively. Note that during the following discussion, we usually omit $\delta$ for the ease of clarification and assume that $\delta$ is fixed. For cpSGD, the known tightest bound is by the combination of standard privacy amplification (Balle et al., 2018) and advanced composition (Dwork et al., 2010). Concretely, if a mechanism costs $\epsilon$ privacy budget, then after composed with sub-sampling the privacy budget is reduced to $\mathcal{O}(\gamma\epsilon)$ where $\gamma$ is the sub-sampling rate. If the mechanism is composed sequentially $T$ times, the privacy budget grows to $\mathcal{O}(\sqrt{T \log(1/\delta)}\epsilon)$. Thus, the total privacy budget of cpSGD is $\mathcal{O}(\gamma\sqrt{T \log(1/\delta)}\epsilon)$. On the other hand, D²P-FED provides a total privacy budget of $\mathcal{O}(\gamma\sqrt{T}\epsilon)$, saving a factor of $\sqrt{\log(1/\delta)}$. Since $\delta$ is typically very small, the saving is quite significant. If the privacy budgets for the two protocols are the same, then D²P-FED can use noise with $\mathcal{O}(\sqrt{\log(1/\delta)})$ smaller scale than cpSGD in each round. This will lead to a $\mathcal{O}(\sqrt{\log(1/\delta)})$-time faster convergence. Then for a given gradient bound, D²P-FED can reach it with $\mathcal{O}(\log(1/\delta))$-time fewer rounds and thus save $\mathcal{O}(\log(1/\delta))$-time communication cost.

Both D²P-FED and cpSGD intrinsically require secure aggregation to establish their privacy guarantee. Agarwal et al. (2018) did not discuss the issue explicitly. As pointed out in Bonawitz et al. (2017), once combined with secure aggregation, each field has to expand at least $\gamma n$ times ($\gamma n$ is the number of chosen clients) to prevent overflow of the sum.

## 7 EVALUATION

We would like to answer the following three questions using empirical evaluation: (1) How D²P-FED performs in a multi-round federated learning compared to cpSGD under either the same privacy guarantee or the same communication cost? (2) How different choices of hyper-parameters affect the performance of D²P-FED? (3) Does D²P-FED work under heterogeneous data distribution? Due to space limitation, we present our main results for (1) in this section and defer the results for (2) and (3) to Appendix E.

### 7.1 EXPERIMENT SETUP

To answer the above questions, we evaluated D²P-FED and cpSGD on INFIMNIST (Bottou (2007)) and CIFAR10 (Krizhevsky et al. (2009)). We sampled 10M hand-written digits from INFIMNIST and randomly split the data among 100K clients. In each round, 100 clients are randomly chosen to upload their difference vectors to train a three-layer MLP. For CIFAR10, we select 10 out of 2000 clients in each round to train a 2-layer convolutional network. All RDP bounds are converted to $(\epsilon, \delta)$-DP for the ease of comparison and the total $\delta$ is set to $1e{-}5$ for all experiments.

### 7.2 MODEL ACCURACY VS. PRIVACY BUDGET

To answer the first question, we studied model accuracy under the same privacy budget as shown in Figure 1a. Compared with cpSGD, D²P-FED achieves 4.7% higher model accuracy on INFIMNIST

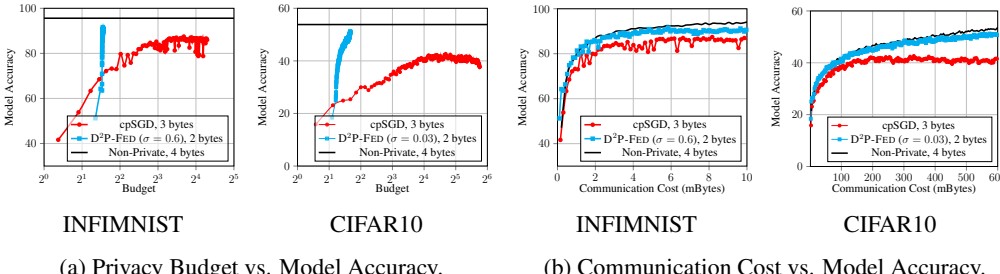

(a) Privacy Budget vs. Model Accuracy.    (b) Communication Cost vs. Model Accuracy.

Figure 1: $D^2$P-FED vs. cpSGD.

and 13.0% higher model accuracy on CIFAR10 after convergence. As expected, $D^2$P-FED composes far tighter under sub-sampling as the lines are much sharper than those of cpSGD. Consequently, $D^2$P-FED converges at a smaller privacy budget than cpSGD as well. Although in Figure 1a cpSGD has better accuracy in the high privacy region, it is not necessarily the case but depends on the scale of the discrete Gaussian noise, as studied in Section E.1. Note that the results for cpSGD in Figure 1 is different from the results in Figure 2 in the original paper (Agarwal et al., 2018). The reason is that in the original paper, they do not sub-sample clients in each round but assign each client to exactly one round beforehand to avoid composition which cpSGD cannot handle well. However, the scheme is far from practical in the real world due to the dynamic nature of the clients.

### 7.3    MODEL ACCURACY VS. COMMUNICATION COST

To answer the second question, we also studied the model accuracy under the same communication cost. As shown in Figure 1b, $D^2$P-FED consistently achieves better model accuracy under the same communication cost on both INFIMNIST and CIFAR10. The main reason is that the tight composition property allows $D^2$P-FED to use smaller per-feature communication cost while still achieving better accuracy. As a concrete instance, $D^2$P-FED with 50% compression rate can achieve better accuracy than cpSGD with 25% compression rate. cpSGD with 50% compression rate either leads to an unacceptable privacy budget or does not converge.

## 8    CONCLUSION

In this work, we developed $D^2$P-FED to achieve both differential privacy and communication efficiency in the context of federated learning. By applying the discrete Gaussian mechanism to the private data transmission, $D^2$P-FED provides stronger privacy guarantee, better composability and smaller communication cost than the only prior work, cpSGD, both theoretically and empirically.

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

## A  PROOF FOR THEOREM 1

We consider the Rényi divergence between two discrete Gaussian distributions differing in the mean value.

*Proof.*

$D_\alpha(N_\mathbb{L}(0, \sigma^2) \| N_\mathbb{L}(\mu, \sigma^2))$

$$\overset{(1)}{=} \frac{1}{\alpha - 1} \log \sum_\mathbb{L} \frac{1}{\sum_\mathbb{L} \exp(-(x-\mu)^2/(2\sigma^2))} \exp(-\alpha x^2/(2\sigma^2)) \cdot \exp(-(1-\alpha)(x-\mu)^2/(2\sigma^2))$$

$$= \frac{1}{\alpha - 1} \log \frac{1}{\sum_\mathbb{L} \exp(-(x-\mu)^2/(2\sigma^2))} \sum_\mathbb{L} \exp((-x^2 + 2(1-\alpha)\mu x - (1-\alpha)\mu^2)/(2\sigma^2))$$

$$\overset{(2)}{\leq} \frac{1}{\alpha - 1} \log \{ \exp((\alpha^2 - \alpha)\mu^2/(2\sigma^2)) \}$$

$$= \alpha \mu^2/(2\sigma^2)$$

$\square$

(1) Because $range(f) \subseteq \mathbb{L}$, we only consider $\mu \in \mathbb{L}$. Thus the denominator of $N_\mathbb{L}(0, \sigma^2)$ and $N_\mathbb{L}(\mu, \sigma^2)$ cancels out as $\sum_\mathbb{L} \exp(-(x-\mu)^2/(2\sigma^2))$ is periodic. (2) $\frac{\sum_\mathbb{L} \exp(-(x-\mu)^2/(2\sigma^2))}{\sum_\mathbb{L} \exp(-(x-\mu)^2/(2\sigma^2))} = \frac{\sqrt{\pi}\vartheta((1-\alpha)\pi\mu, e^{-\pi^2})}{\vartheta(0, \frac{1}{e})} \leq 1$ where $\vartheta$ is the Jacobi theta function (Wikipedia).

## B  PROOF FOR THEOREM 2

*Proof.* The $l_2$ sensitivity of $\delta$ is naturally bounded by $2D$. The rotation does not change the sensitivity. The $k$-level quantization might expand the space as shown in Figure 2. An upper bound on the radius of the red circle is $D + \sqrt{d}\frac{D}{k-1}$. When we take $k = \sqrt{d} - 1$, it reduces to $2D$. Thus, the upper bound on the sensitivity is $4D$. $\square$

## C  PROOF FOR THEOREM 3

We first prove the following lemma.

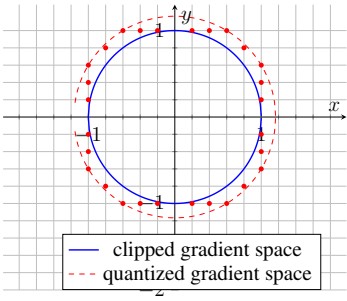

Figure 2: Sensitivity after Quantization.

**Lemma 3.**

$$\sum \phi_q(x) = \phi_q(\sum x)$$

*Proof.*

$$
\begin{aligned}
\sum \phi_q(x) &= \sum \frac{2\boldsymbol{g}^{\max}}{k-1}((\frac{k-1}{2\boldsymbol{g}^{\max}}x + \frac{q-1}{2}) \mod q - \frac{q-1}{2}) \\
&= \frac{2\boldsymbol{g}^{\max}}{k-1}((\frac{k-1}{2\boldsymbol{g}^{\max}}\sum x + \frac{q-1}{2}) \mod q - \frac{q-1}{2}) \\
&= \phi_q(\sum x)
\end{aligned}
$$

$\square$

Then we prove Theorem 3 as follows.

*Proof.* Given Lemma 3,

$$
\begin{aligned}
\tilde{\boldsymbol{g}}_t &= \frac{1}{k} \sum_{i \in S} \tilde{\boldsymbol{g}}_t^{(i)} \\
&= \frac{1}{k} \sum_{i \in S} \phi_q(\phi_q(\tilde{\boldsymbol{g}}_{t,dp}^{(i)}) + \sum_{j \neq i, j \in S} \boldsymbol{u}_{ij} - \sum_{j \neq i, j \in S} \boldsymbol{u}_{ji}) \\
&= \frac{1}{k} \phi_q(\sum_{i \in S} (\phi_q(\tilde{\boldsymbol{g}}_{t,dp}^{(i)}) + \sum_{j \neq i, j \in S} \boldsymbol{u}_{ij} - \sum_{j \neq i, j \in S} \boldsymbol{u}_{ji})) \\
&= \frac{1}{k} \phi_q(\sum_{i \in S} \tilde{\boldsymbol{g}}_{t,dp}^{(i)})
\end{aligned}
$$

The expression $\sum_{i \in S} \hat{\boldsymbol{g}}_{t,dp}^{(i)}$ forms a centralized discrete Gaussian mechanism and according to the post-processing theorem, the same RDP guarantee still holds. $\square$

## D    PROOF FOR THEOREM 4

*Proof Sketch.* Before starting the proof, we introduce two lemmas for the proof.

**Lemma 4** (Proposition 19 from Canonne et al. (2020))**.** *For all $\sigma \in \mathbb{R}$ with $\sigma > 0$,*

$$\mathbb{V}[N_{\mathbb{Z}}(0, \sigma^2)] \leq \sigma^2(1 - \frac{4\pi^2\sigma^2}{e^{4\pi^2\sigma^2-1}}) < \sigma^2$$

*Moreover, if $\sigma^2 \leq \frac{1}{3}$, then $\mathbb{V}[N_{\mathbb{Z}}(0, \sigma^2)] \leq 3 \cdot e^{-\frac{1}{2\sigma^2}}$*

**Lemma 5** (Proposition 23 from Canonne et al. (2020)). *For all $m \in \mathbb{Z}$ with $m \geq 1$ and all $\sigma \in \mathbb{R}$ with $\sigma > 0$,*

$$\mathbb{P}_{X \sim N_{\mathbb{Z}}(0,\sigma^2)}[X \geq m] \leq \mathbb{P}_{X \sim N(0,\sigma^2)}[X \geq m-1].$$

*Moreover, if $\sigma \geq 1/\sqrt{2\pi}$, we have*

$$\mathbb{P}_{X \sim N_{\mathbb{Z}}(0,\sigma^2)}[X \geq m] \geq \frac{1}{1 + 3e^{-2\pi^2\sigma^2}} \mathbb{P}_{X \sim N(0,\sigma^2)}[X \geq m]$$

Now we start our proof. MSE can be rewritten in the following format.

$$\mathbb{E}[\|\hat{\bar{X}} - \bar{X}\|_2^2] = \frac{1}{n^2} \sum_{j=1}^{d} \sum_{i=1}^{n} \mathbb{E}[(\hat{\bar{X}}_i(j) - X_i(j))^2]$$

For each $\star = \mathbb{E}[(\hat{\bar{X}}_i(j) - X_i(j))^2]$, we need to consider two cases. If no overflow happens, due to Lemma 4

$$\star_{\neg o} \leq \mathbb{E}[(\frac{2X^{\max}}{k-1})^2(\mathbb{V}(Ber(p_i(j))) + \mathbb{V}(N_{\mathbb{L}}(\sigma)))] \leq \frac{4(X^{\max})^2}{(k-1)^2}(\frac{1}{4} + \frac{\sigma^2}{\gamma^2 n^2})$$

If overflow happens, trivially we have $\star_o \leq k^2$. Thus, we have

$$\star = \mathbb{P}[\neg o] \cdot \star_{\neg o} + \mathbb{P}[o] \cdot \star_o \leq \mathbb{P}_{X \sim N_{\mathbb{L}}(\sigma^2)}[X \leq q] \cdot \star_{\neg o} + \mathbb{P}_{X \sim N_{\mathbb{L}}(\sigma^2)}[X \geq q-k] \cdot \star_o \quad (1)$$

With Lemma 5, we can bound the probabilities and get the final MSE result in the theorem.

$$\square$$

# E   OTHER EVALUATION

## E.1   INFLUENCE OF NOISE SCALE

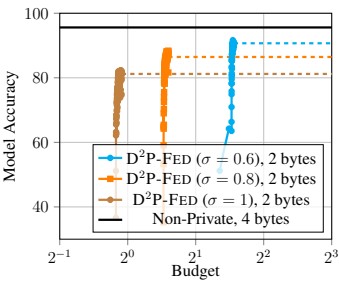

Figure 3: $D^2$P-FED under different $\sigma$.

The hyper-parameter of the most vital interest in $D^2$P-FED is the scale of the noise. To understand the effect of the noise scale, we evaluated $D^2$P-FED on INFIMNIST, with 3 different choices of noise scale as shown in Figure 3. It is no surprise that the higher the noise scale, the smaller the privacy budget and the lower the model accuracy. This also illustrates the claim we have in Section 7.2 that $D^2$P-FED can also have relatively good performance in the high privacy region at the cost of the model accuracy.

## E.2   INFLUENCE OF HETEROGENEOUS DATA DISTRIBUTION

It is well known that data is sometimes heterogeneously distributed among clients in a federated learning system. Thus, to better understand $D^2$P-FED's behavior under heterogeneous data distribution, we simulated heterogenenous data distribution by distributing the INFIMNIST data according to the classes and evaluated $D^2$P-FED on these clients. The results are shown in Figure 4 and we can see that under heterogeneous data distribution the model accuracy drops by more than 10%. This complies with the previous empirical results and there have been a line of researches focusing on addressing the issue (Yurochkin et al., 2019a;b; Wang et al., 2020). Although orthogonal to this paper, we deem it as an interesting open problem how to integrate these works with $D^2$P-FED.

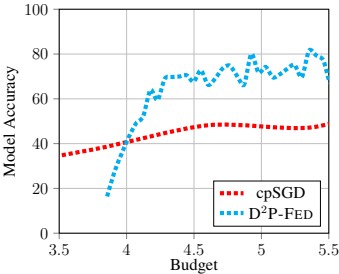

Figure 4: $D^2$P-FED under homogeneous/heterogeneous distribution.

### E.3 INFLUENCE OF GROUP SIZE $q$

We also ran $D^2$P-FED with multiple choices of discrete group size $q$. We observe that once the noise scale $\sigma$ is fixed, the performance is relatively robust to $q$.

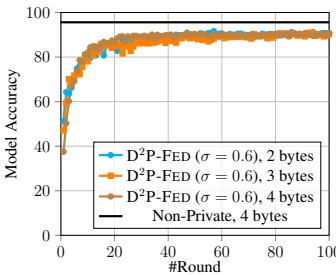

Figure 5: $D^2$P-FED under different group size $q$.

