# OpenReview forum: "D2p-fed:Differentially Private Federated Learning with Efficient Communication"
_ICLR.cc/2021/Conference — Reject_

### Official Review · AnonReviewer2 · 2020-10-19
**Some deficiencies**

**Rating:** 4
**Confidence:** 3

**Review:**

The authors propose D2P-FED to achieve differential privacy and communication efficiency in FL tasks. The framework is interesting and the problem studied is important. However, this paper suffers from some major deficiencies.

Major comments:

1) It is claimed that the privacy guarantee for D2P-FED is better than that of cpSGD. However, this is not substantiated after the privacy results of Theorem 5. Furthermore, the bound here depends on D, the clipping threshold on the l_2 norm of g, which clearly affects the privacy guarantee per Theorem 5.

2) The algorithm here depends on the lattice being defined as 2g^{max}/(k-1) Z. It is not clear how one can set the g^{max} in practice. The differences of the parameters w_t are generated on the fly for each client. Thus, this algorithm has limited applicability in practice.

3) Similarly to point 1), it is not clear in what way the communication cost is better than that of cpSGD. From the convergence result in Theorem 7, it seems like there is no improvement. Further, g^{max} needs to be set in a very specific way that is dependent on D and d, which is again not known in practice. So whether we can attain the convergence rate is unclear.

4) The authors acknowledge in their experiments that the results are sensitive to the scale of the discrete Gaussian noise. Hence, the privacy-communication cost-convergence rate tradeoff is not clearly delineated.

Minor comments:

1) The writing is not sufficiently precise and clear. For example, before the statement of Thm 1, it is mentioend that the proof is delayed to Appendix A. However, the proof of Thm 1 is not provided therein. Rather the proof of Thm 3 is in that appendix. The proof of Thm 4 in Appendix B is not sufficiently precise. It is not clear what the authors mean by "expand the space a little bit". How much is "a little bit"? Since this is a "theorem", the proof should be watertight.

---

> ### Author Response · Authors · 2020-11-15
> **Response to Reviewer 2 (Part 1)**
>
> We would like to thank the reviewer for the constructive and detailed comments. Please see below for our response.
>
> Q: It is claimed that the privacy guarantee for D2P-FED is better than that of cpSGD. However, this is not substantiated after the privacy results of Theorem 5 (Corollary 3 in the latest version). Furthermore, the bound here depends on $D$, the clipping threshold on the l_2 norm of g, which clearly affects the privacy guarantee per Theorem 5 (Corollary 3 in the latest version).
>
> A: Thanks for the review but there seems to be a misunderstanding about the statement due to our insufficient elaboration. First, almost all DP mechanisms for non-convex optimization are based on gradient clipping and further add noise proportional to gradient norm bound. The privacy guarantee of such mechanisms cannot get away with a dependence on the gradient norm bound $D$. For instance, in the groundbreaking work [1] and [2], the privacy bounds also implicitly linearly depend on $D$ (Theorem 1 in [1] and Corollary 3 in [2]). $D$ does not explicitly show in the bounds because they assume that $D=1$ (Lemma 3 in [1] and Corollary 3 in [2]). Furthermore, cpSGD’s privacy guarantee also depends linearly on $D$ (Theorem 1 and Equation (11)). Therefore, we do not consider the dependence on $D$ a defect of our approach. Second, the claim that D2P-FED has a better privacy guarantee than cpSGD can be mainly justified by the following four aspects:
> (1) D2P-FED can achieve a stronger privacy notion than cpSGD. Specifically, D2P-FED can achieve RDP yet cpSGD can only guarantee ($\epsilon, \delta$)-DP and it is proved in [2] that RDP is a strictly stronger notion than ($\epsilon, \delta$)-DP. ($\epsilon, \delta$)-DP allows a small probability (on the order of $\delta$) of complete privacy failure. For example, in cpSGD, given two neighboring datasets (e.g. $D_1$ and $D_2$) differing in one entry, if the outputs of the learning algorithm on the two datasets differ by $1$ (e.g. $0$, $1$), then after adding noise drawn from the same binomial distribution (e.g. $Binom(m, p)$), the supports of outputs also differ by $1$ (e.g. {$0, \cdots, m$}, {$1, \cdots, m+1$}). Therefore, if the output is $0$, we can immediately tell that the input is $D_1$. On the other hand, as demonstrated in [2], RDP’s privacy guarantee “degrades gracefully, such as to $1$-DP with probability $\frac{\delta}{2}$, to $2$-DP with probability $\frac{\delta}{4}$, etc”.
> (2) D2P-FED enjoys a tighter composition compared to cpSGD. The best way so far to derive the total privacy budget for multiple rounds for cpSGD is to use advanced composition theorem, as cpSGD follows ($\epsilon, \delta$)-DP. On the other hand, since D2P-FED follows RDP, we can leverage analytical moments accountant [3] to calculate the total privacy budget. As shown in [3], advanced composition in general produces much looser composition than analytical moments accountant. The reason is that if we use advanced composition for multiple-round composition, the total privacy budget has a complex dependency on the per-round privacy budget and $\delta$; hence, it is often computationally hard to find the optimal $\epsilon$ given a fixed $\delta$. By contrast, by tracking the cumulant generating function of the composed mechanisms symbolically, analytical moments accountant allows a convenient conversion from RDP bound to ($\epsilon, \delta$)-DP with the smallest $\epsilon$ given fixed $\delta$.
> (3) Our experimental results in Figure 1 also empirically show that D2P-FED enjoys a tighter composition than cpSGD. The total privacy budget for D2P-FED grows much more slowly than cpSGD as training proceeds.
> (4) Moreover, according to Figure 1 in [4], even with the same noise scale, Gaussian noise provides stronger privacy guarantee than Binomial noise. As discrete Gaussian noise follows the same RDP bound as Gaussian noise, we believe discrete Gaussian can map the same-scale noise to lower privacy cost.
>
> We have added more details after Corollary 3 about the privacy guarantee comparison between D2P-FED and cpSGD.

---

> > ### Author Response · Authors · 2020-11-15
> > **Response to Reviewer 2 (Part 2)**
> >
> > Q: The algorithm here depends on the lattice being defined as $\frac{2g^{max}}{k-1} \mathbb{Z}$. It is not clear how one can set the $g^{max}$ in practice. The differences of the parameters $w_t$ are generated on the fly for each client. Thus, this algorithm has limited applicability in practice.
> >
> > A: Thanks for the comment! The $g^{max}$ parameter is not a parameter unique to our approach. Indeed, it is inherited from cpSGD and is necessary for differential privacy to hold in FL with quantization. $g^{max}$ does not need to be chosen on the fly. Since the clients clip their gradients before quantization, a natural choice of $g^{max}$ is the clipping bound D. If we want to set $g^{max}$ as $\sqrt{\frac{\log d}{d}}D$ as claimed in the remark under Theorem 4, we can require the clients to clip the $\ell_\infty$ norm of the gradient after random rotation. Therefore, we do not agree that $g^{max}$ affects the applicability in practice. We have emphasized the choice of $g^{max}$ in the newest revision in the remark under Theorem 4.
> >
> > Q: Similarly to point 1), it is not clear in what way the communication cost is better than that of cpSGD. From the convergence result in Theorem 7 (Theorem 4 in the latest revision), it seems like there is no improvement. Further, $g^{max}$ needs to be set in a very specific way that is dependent on $D$ and $d$, which is again not known in practice. So whether we can attain the convergence rate is unclear.
> >
> > A: Thanks for the review but there seems to be a misunderstanding about the statement due to our insufficient elaboration. (1) Given a fixed privacy budget, due to the tighter composition of sub-sampled RDP [3], each round of D2P-FED can have more privacy budget and thus smaller noise scale $\sigma$. Plugging a smaller $\sigma$ into Theorem 4 will give a better MSE and thus a better convergence rate. The way to choose $g^{max}$ is provided in the previous answer and updated in the newest revision. (2) Given the better convergence rate, we can achieve the same accuracy with a smaller number of rounds. Since our per-round communication is the same as cpSGD by Theorem 5, our total communication cost is smaller than cpSGD. This is also illustrated by our empirical evaluation (Figure 1(b)). We have added a remark to better demonstrate the advantages under Theorem 4 in the newest revision.
> >
> > Q: The authors acknowledge in their experiments that the results are sensitive to the scale of the discrete Gaussian noise. Hence, the privacy-communication cost-convergence rate tradeoff is not clearly delineated.
> >
> > A: Thanks for the helpful comment! The sensitivity to noise scale is not unique to our approach. Indeed, the Gaussian mechanism applied to SGD (the most common way to achieve DP in SGD) [1] as well as cpSGD both have a noise scale as one of the free parameters. In existing works [1,2,3], the selection of noise scale is often not much discussed and fixed to some value that works for respective prediction tasks. In our work, we run multiple noise scales for our method and cpSGD, and show the result corresponding to the noise scale that can achieve best performance within a meaningful privacy range. We believe that this is a fair comparison. We apologize that we are not as clear as we intended about the tradeoff among privacy, communication cost, and convergence trade. Actually, this tradeoff can be directly read off from Figure 1. For a given reasonable privacy budget or a communication budget, our method can achieve higher model accuracy than the baseline. We’d appreciate it if the reviewer has further suggestions.

---

> > > ### Author Response · Authors · 2020-11-15
> > > **Response to Reviewer 2 (Part 3)**
> > >
> > > Q: The writing is not sufficiently precise and clear. For example, before the statement of Thm 1, it is mentioned that the proof is delayed to Appendix A. However, the proof of Thm 1 is not provided therein. Rather the proof of Thm 3 is in that appendix. The proof of Thm 4 in Appendix B is not sufficiently precise. It is not clear what the authors mean by "expand the space a little bit". How much is "a little bit"? Since this is a "theorem", the proof should be watertight.
> > >
> > > A: Thanks for pointing these out! (1) The proof in appendix A is actually for Thm 1 but the title is wrong. We have corrected the typo in the newest revision. (2) We have refined the wording to make the proof more clear and rigorous. We would appreciate it if the reviewer has further suggestions.
> > >
> > > [1] Abadi, Martin, et al. "Deep learning with differential privacy." Proceedings of the 2016 ACM SIGSAC Conference on Computer and Communications Security. 2016.
> > > [2] Mironov, Ilya. "Rényi differential privacy." 2017 IEEE 30th Computer Security Foundations Symposium (CSF). IEEE, 2017.
> > > [3] Wang, Yu-Xiang, Borja Balle, and Shiva Prasad Kasiviswanathan. "Subsampled Rényi differential privacy and analytical moments accountant." The 22nd International Conference on Artificial Intelligence and Statistics. PMLR, 2019.
> > > [4] Agarwal, Naman, et al. "cpsgd: Communication-efficient and differentially-private distributed sgd." Advances in Neural Information Processing Systems. 2018.

---

### Official Review · AnonReviewer3 · 2020-10-27

**Rating:** 7
**Confidence:** 4

**Review:**

##########################################################################

Summary:


The paper proposes the discrete Gaussian based differentially private federated learning algorithm to achieve both differential privacy and communication efficiency in federated learning. In particular, it adds discrete Gaussian noise into client updates and uses secure aggregation to prevent the server from observing the individual updates. The algorithm satisfies RDP and has lower communication cost compared to the previous method cpSGD.

##########################################################################

Reasons for score:

 I like this work. 1. The problem is critical in federated learning. The D2P-Fed algorithm has nice performance on the trade-off among privacy, utility, and communication cost. 2. It has several algorithmic novelties. To employ secure aggregate, it's natural to use a mechanism with discrete support (or discretize that). The discrete Gaussian mechanism is indeed a better candidate than the binomial mechanism. But there is much more beyond a simple combination of the discrete Gaussian mechanism and secure aggregation. They use several other techniques, such as stochastic quantization and random rotation. The mapping to a cyclic additive group is novel. 3. They provide strong theoretical guarantees and empirical validation.

##########################################################################
Minor comments:

1. In Section 3 under FL Overview, I don't think $k$ is clearly defined. I guess $k=\gamma n$. It is also confusing in Algorithm 1. It should be clearly defined. The same for $d$ in Algorithm 1.

2. The probability $g[j]-b[r]/b[r+1]-b[r]$ doesn't seem make sense. Like if $g[j]=b[r+1]$, it will be set to $b[r]$ with probability 1. Should it be $1-(g[j]-b[r]/b[r+1]-b[r])$?

3. In Figure 1(b), the x-axis should be communication cost for CIFAR10.

---

> ### Author Response · Authors · 2020-11-15
> **Response to Reviewer 3**
>
> Thanks for your positive feedback on our paper! We are glad that you like the paper and find your comments really helpful. Please see below for our response.
>
> Q: In Section 3 under FL Overview, I don't think $k$ is clearly defined. I guess $k=\gamma n$. It is also confusing in Algorithm 1. It should be clearly defined. The same for $d$ in Algorithm 1. The probability $\frac{g[j]−b[r]}{b[r+1]−b[r]}$ doesn't seem make sense. Like if $g[j]=b[r+1]$, it will be set to $b[r]$ with probability $1$. Should it be $1−\frac{g[j]−b[r]}{b[r+1]−b[r]}$? In Figure 1(b), the x-axis should be communication cost for CIFAR10.
>
> A: Thanks for pointing this out. We have corrected the notation problems and the typos in Algorithm 1 in the latest revision. We would appreciate it if the reviewer has further suggestions.

---

### Official Review · AnonReviewer4 · 2020-10-28
**This paper proposed a privacy-preserving federated learning algorithm with efficient communication.**

**Rating:** 6
**Confidence:** 4

**Review:**

This paper proposed a privacy-preserving federated learning algorithm with efficient communication. To save communication costs, they integrate stochastic quantization and random rotation. Moreover, they apply the discrete Gaussian mechanism with Renyi DP analysis for privacy guarantee.


Novelty & Significance
-----------------------------

Solving federated learning with a privacy guarantee is of increasing practical importance, and certainly trying to do so with efficient communication efficiency is more important.

The RDP analysis for discrete Gaussian mechanism is nice, which allows a tighter composition with analytical moments accountants.


Technical concern:
-----------------------------

 I have concerns regarding the claim ''D2P-FED is more communication efficient compared to cpSGD'' due to a $log(1/\delta)$ factor in big O. Note that constant matters in differential privacy. Without knowing other constants hidden in big O, I feel hard to follow the conclusion there.


Experimental Evaluation:
----------------------------

My major concern is that empirical evaluation is limited.
(1) In Figure 1, three methods use different quantization. There lacks clarification on the choice of bytes.  Moreover, it would be much better to add a non-private baseline without quantization.
(2) Noting the privacy budget greater than 10 is no longer meaningful; it may be better to focus on the practical privacy regimes in Figure 1 (a).
(3) In Figure 1 (b), is the privacy budget aligned? If not, how to set the noise scale for cpSGD and D2P-FED?  Is it possible to align both the privacy budget and communication cost? For example, the per-round communication is the same for both algorithms with the same bytes. Set the x-axis to be the communication cost, the privacy cost is fixed for all points, and the y-axis reports the model accuracy.
(4) INFIMNIST and CIFAR10 are still two toy datasets. Perhaps authors can add more experiments with real-world datasets in the next version.

---

> ### Author Response · Authors · 2020-11-15
> **Response to Reviewer 4 (Part 1)**
>
> We thank the reviewer for the helpful comments. Please see below for our response.
>
> Q: I have concerns regarding the claim ''D2P-FED is more communication efficient compared to cpSGD'' due to a $log(1/\delta)$ factor in big $\mathcal{O}$. Note that constant matters in differential privacy. Without knowing other constants hidden in big $\mathcal{O}$, I feel hard to follow the conclusion there.
>
> A: Thanks for the question! We totally agree that constant matters in DP. The $log(1/\delta)$ factor indicates the communication efficiency of D2P-FED in theory but cannot serve as a determining evidence. However, tight constant analysis is extremely hard (even impossible) in most cases. Therefore we choose to demonstrate the communication efficiency using empirical evaluation just like what most relevant works do. We believe that Figure 1(b) empirically illustrates the communication efficiency of D2P-FED over cpSGD.

---

> > ### Author Response · Authors · 2020-11-15
> > **Response to Reviewer 4 (Part 2)**
> >
> > Q: My major concern is that empirical evaluation is limited. (1) In Figure 1, three methods use different quantization. There lacks clarification on the choice of bytes. Moreover, it would be much better to add a non-private baseline without quantization. (2) Noting the privacy budget greater than 10 is no longer meaningful; it may be better to focus on the practical privacy regimes in Figure 1 (a). (3) In Figure 1 (b), is the privacy budget aligned? If not, how to set the noise scale for cpSGD and D2P-FED? Is it possible to align both the privacy budget and communication cost? For example, the per-round communication is the same for both algorithms with the same bytes. Set the x-axis to be the communication cost, the privacy cost is fixed for all points, and the y-axis reports the model accuracy. (4) INFIMNIST and CIFAR10 are still two toy datasets. Perhaps authors can add more experiments with real-world datasets in the next version.
> >
> > A: Thanks for the detailed suggestions! We answer the questions one by one as below.
> >
> > (1) Actually we ran both cpSGD and D2P-FED with three different quantization schemes: 2bytes, 3bytes, 4bytes. We tested integer multiples of 1byte because it is the smallest network transmission unit. For D2P-FED, we exhibited the result only for 2 bytes because the results for 3 bytes and 4 bytes are basically the same as 2 bytes. We have added the results for 3 and 4 bytes in Appendix E in the newest revision. For cpSGD, we cannot obtain meaningful results with 2-byte quantization after one-week tuning: either the privacy budget skyrockets or the model accuracy drops drastically. One potential reason is that cpSGD’s privacy budget is approximately proportional to $\frac{1}{\sqrt{k}}$, where $k$ is the quantization level. Besides, $k$ is also directly related to communication cost in cpSGD (which is not the case in D2P-FED whose communication cost is decided by group size $q$). Thus, 2-byte quantization leads to $2^8=256$ times greater privacy budget than 3-byte quantization in cpSGD if all the other hyper-parameters are fixed. In contrast, the privacy budget in D2P-FED does not rely on the quantization level and thus is not strongly affected by quantization. Besides, we have added the baseline case for Figure 1(b) in the newest revision.
> >
> > (2) We agree that the low privacy regime (e.g. $\epsilon \geq 10$) does not provide a strong privacy guarantee. We include the low privacy regime into our plot because we would like to compare the privacy budget for cpSGD and D2P-FED with the same training iterations. While the privacy budget of D2P-FED stays below $4$, the privacy budget of cpSGD has entered this low privacy regime. This confirms the insufficiency of cpSGD under composition from the empirical perspective.
> >
> > (3) If we understand correctly, “aligned privacy budget” means the same privacy budget for each round (please correct us if we are wrong). If so, the privacy budget is not aligned in our experiments. The main reason is that D2P-FED leverages analytical moments accountant [1] to obtain the smallest $\epsilon$ for any given fixed $\delta$. Analytical moments accountant is a dynamic data structure and there is no closed-form representation for the total privacy budget of a composite mechanism. We cannot analytically derive the per-round privacy budget from the total budget and fix it beforehand. Moreover, cpSGD’s communication cost and privacy cost cannot be independently controlled as they both depend on the number of trials in the Binomial distribution (i.e., $m$ in $Binom(m, p)$). Hence, it is challenging to align privacy budgets or align privacy budgets and communication cost simultaneously. As for the noise scale, we run experiments for multiple noise scales for cpSGD and D2P-FED. The results we exhibited for the two methods in the paper are based on the respective noise scale that can achieve highest accuracy within a meaningful privacy budget regime.
> >
> > (4) We selected INFIMNIST and CIFAR10 for evaluation as these are the most commonly used ones for studying differentially private learning. Indeed, cpSGD is only evaluated on INFIMNIST. However, we are happy to add more experiments with larger datasets into the next revision. We have started experiments on larger datasets following the setting in [2]. We will report the results as soon as we obtain them.
> >
> > We will appreciate it if the reviewer has further suggestions.
> >
> > [1] Wang, Yu-Xiang, Borja Balle, and Shiva Prasad Kasiviswanathan. "Subsampled Rényi differential privacy and analytical moments accountant." The 22nd International Conference on Artificial Intelligence and Statistics. PMLR, 2019.
> > [2] Rothchild, Daniel, et al. "FetchSGD: Communication-Efficient Federated Learning with Sketching." Proceedings of Machine Learning Research (2020).

---

### Official Review · AnonReviewer1 · 2020-10-30
**An interesting idea, but it could be presented more clearly and more convincingly.**

**Rating:** 5
**Confidence:** 4

**Review:**



This paper proposes to use a discrete gaussian distribution the generate noise for differential privacy.
While it is claimed that the discrete Gaussian distribution is better than the binomial distribution, no in-depth comparison is made.  The discrete Gaussian distribution, a discretized version of the Gaussian distribution, works as expected, and provides all properties the Gaussian distribution provides (if the discretization is sufficiently fine-grained).
An advantage is that the communication cost can be reduced by stochastic quantization of the values to be transmitted.  It is unclear whether the same stochastic quantization technique can be used for the classic Gaussian distribution.
The ideas of discretizing, of stochastic quantization to reduce communication cost, and the use of discrete distributions for differential privacy are all already known, the (limited) novelty is in the proper combination of these ideas.
The text is understandable for readers who know already the meaning of all symbols and names used, but a broader audience could be reached by properly introducing all concepts and notations in an understandable, coherent and self-contained way.  This would be valuable, even if it would mean that some more technical elements would need to be moved to an appendix.


In conclusion, while the idea is interesting, I have some concerns with the presentation and the insufficiently motivated novelty (and the related lack of more in-depth comparison with existing work).


* The paper points to a few shortcomings in the literature, but exaggerates their importance.  E.g., when the text says "However, cpSGD faces several limitations when applied to real-world applications. Firstly, with Binomial noise, the output of a learning algorithm would have different supports when any client participates or withdraws from learning", it may be true this is the case when literally considering the cpSGD paper, but such problem typically can be overcome easily.  When the text concludes "can only guarantee approximate DP where the participation of the client can be completely exposed with nonzero probability." I guess "approximate DP" means (epsilon,delta)-differential privacy where delta is strictly larger than zero (which is usually anyway already the case for Gaussian and binomial mechanisms).

* Definition 3: I wonder why the numerator and denominator of the exponent in the probability e^{-\pi x^2/(2\pi\sigma^2)} both contain \pi and this fraction isn't simplified.
* Definition 3: please clarify what kind of "discrete additive subgroup" you have in mind: is it a finite group (where addition is possibly modulo the group order) or is it a multiple of the set of integers \mathbb{Z} ? (much later in the paper, it becomes increasingly clear that the latter option is the correct one)
* Section 5.1: please explain the meaning of all used variables at their first use. E.g., the k in an expression log(k) isn't explained, until Section 6.1 cites another work for the meaning of k-level quantization (without even explaining the gist of quantization idea briefly).

---

> ### Author Response · Authors · 2020-11-15
> **Response to Reviewer 1**
>
> We thank the reviewer for the constructive comments. Please see below for our response.
>
> Q: The paper points to a few shortcomings in the literature, but exaggerates their importance. E.g., when the text says "However, cpSGD faces several limitations when applied to real-world applications. Firstly, with Binomial noise, the output of a learning algorithm would have different supports when any client participates or withdraws from learning", it may be true this is the case when literally considering the cpSGD paper, but such problem typically can be overcome easily. When the text concludes "can only guarantee approximate DP where the participation of the client can be completely exposed with nonzero probability." I guess "approximate DP" means ($\epsilon,\delta$)-differential privacy where delta is strictly larger than zero (which is usually anyway already the case for Gaussian and binomial mechanisms).
>
> A: Thanks for the review but there seems to be a misunderstanding about the statement due to our insufficient elaboration. We did not intend to use the two cited sentences (regarding changing support and approximate DP (aka $\epsilon,\delta$)-DP) for stating two separate problems of cpSGD. Indeed, the problem of cpSGD lies in the second sentence only (i.e., approximate DP) and the first sentence (i.e., changing support) is for specifying the cause of the problem. We’d like to first clarify that the Gaussian mechanism mentioned by the reviewer can indeed achieve a stronger privacy than approximate DP, which is RDP. So naturally, one would ask: Can RDP be achieved for federated learning with constrained communication? Actually, cp-SGD cannot get away with approximate DP in its original form because of the limited support of Binomial noise (we will provide an example to illustrate the intuition for this later). Besides, we want to emphasize that the problem of changing supports is **not easily solvable**. We agree with the reviewer that there are several potential methods to address the issue, like clipping the noised value into a fixed support. However, these methods will introduce extra sophistication into the privacy analysis and to our best knowledge, there has not been any work that is able to prove that simple renormalization into the fixed support would lead to stronger privacy than approximate DP.
>
> To help understand why the mechanism with varying support for different input can only satisfy approximate DP, we provide the following concrete example:
>
> Assume we have two neighboring datasets $D_1$ and $D_2$ differing in only one row. Given a learning algorithm, the update vector (for clarity we assume this is one-dimensional) for $D_1$ is $0$ while the update vector for $D_2$ is $1$. Then we add binomial noise $Binom(m,p)$ to the two outputs (for clarity we assume we directly add the noise without any shifting and zooming). Thus, the output support for $D_1$ is ${0, 1, \cdots, m}$ while the support for $D_2$ is $\{1, 2, \cdots, m+1\}$. If the output is $0$, then we immediately know that the input is $D_1$ not $D_2$. If we think about the above limitation theoretically, this actually means that the denominator in the pure-DP definition $\epsilon=\log\frac{Pr[A(D1)\in S]}{Pr[A(D2)\in S]}$ can be zero and thus breaks any finite $\epsilon$-DP guarantee with a small probability. Therefore, Binomial mechanism is intrinsically restricted to approximate DP (($\epsilon, \delta$)-DP) which allows a small (but not negligible) probability of complete privacy failure and cannot follow stronger definitions such as RDP.
>
> We have rewritten this part to make it more clear in the latest revision. We would appreciate it if the reviewer has further suggestions.
>
>
> Q: Definition 3: I wonder why the numerator and denominator of the exponent in the probability $e^{-\pi x^2/(2\pi\sigma^2)}$ both contain $\pi$ and this fraction isn't simplified.
>
> A: Thanks for pointing this out. We have simplified it in the newest revision.
>
> Q: Definition 3: please clarify what kind of "discrete additive subgroup" you have in mind: is it a finite group (where addition is possibly modulo the group order) or is it a multiple of the set of integers $\mathbb{Z}$ ? (much later in the paper, it becomes increasingly clear that the latter option is the correct one)
>
> A: In Definition 3, “discrete additive group means” means the latter one. We have changed the wording in definition 3 to make it more clear.
>
> Q: Section 5.1: please explain the meaning of all used variables at their first use. E.g., the $k$ in an expression $\log(k)$ isn't explained, until Section 6.1 cites another work for the meaning of $k$-level quantization (without even explaining the gist of quantization idea briefly).
>
> A: Thanks for pointing this out. We have added explanation for the notations in the latest revision.

---

### Decision · Program_Chairs · 2021-01-07
**Final Decision**

**Decision:**

Reject

**Comment:**

The paper considers differentially private federated learning --- a well-motivated problem. The proposed algorithm is a simple modification to existing methods, e.g., DP-FedAvg, but uses a different DP mechanism for noise-adding.  The reviewers liked the motivation but criticized the work for its incremental nature and for somewhat overselling the contributions.

Pros:

- The paper used advanced Renyi DP accounting to get a stronger privacy-utility tradeoff.

- The experimental results improve over cp-sgd that uses Binomial mechanisms

Cons:

- It is a bit incremental in its contribution.  The main contribution is to applying "discrete Gaussian mechanism" to the federated learning problem for the interest of reducing the communication cost.   Discrete Gaussian mechanism and its RDP analysis are both from existing work.

- The improvement in privacy-utility tradeoff over cp-sgd seems to be due to that the discrete Gaussian mechanism has an RDP bound, which plugs right into the subsampling bound and moments accountant.    It is unclear whether the improvement is coming from the different noise or a stronger privacy accounting.  Notice that the privacy accounting of Binomial mechanism in the initial cp-sgd paper was rather crude, thus a fair comparison would be to also conduct an RDP analysis for the Binomial mechanism.

Overall, there weren't sufficient support among the reviewers and the experimental results alone are not so groundbreakingly strong to carry the paper single-handedly.